# Community knowledge, attitude and practices to SARS-CoV-2 disease 2019 (COVID-19): A cross-sectional study in Woldia town, Northeast Ethiopia

Kindu Alem Molla[ID]*, Silamlak Birhanu Abegaz[ID]

Natural and Computational Sciences Faculty, Department of Biology, Woldia University, Woldia, Ethiopia

* kindualem@wldu.edu.et

**Data Availability Statement:** All data are available in the paper.

## Abstract

SARS-CoV-2 disease 2019 (COVID-19) is pandemic and currently becomes a serious cause of death worldwide. It is caused by a SARS-CoV-2 belonging to a family known as corona virus. The aim of this study is to assess the community knowledge, attitude and practice strategy implementation on SARS-CoV-2 disease 2019 (COVID-19). A cross-sectional survey study was done from July to October, 2020 in Woldia town, Northeast Ethiopia. Interviewer-administered questionnaire was used to collect data from 404 respondents. Data collected were analyzed using descriptive statistics and chi-square test with a 95% confidence interval to know the association of socio-demographic characteristics with the knowledge, attitude and practices towards COVID-19. From a total of 404 responses collected (64.1%, n = 259/404), (50.7%, n = 205/404) and (39.6%, n = 160/404) of the respondents were between ages 18–39 years, males and were diploma and above, respectively. The majority of the respondents had good knowledge about the transmission mode and symptoms of COVID-19 and they obtained information mainly through mass media. The knowledge of the respondents about the transmission mode of COVID-I9 through coughing and sneezing, direct contact with infected person and touching contaminated materials was statistically associated with education and occupation (p <0.001). Among the total respondents (53.7%, n = 217/404) had a negative attitude that COVID-19 pandemic will not be controlled. The attitude of the respondents towards successfully controlling of COVID-I9 was statistically associated in terms of age, marital status, education and occupation (p < 0.001). Most of the respondents (63.1%, n = 255/404), (58.9%, n = 238/404), (66.8%, n = 270/404) and (63.9%, n = 258/404) did not wash hands with soap, avoidance of touching the nose and mouth, practicing social distancing and wearing of face masks in public or crowded places, respectively. The practices of the respondents towards COVID-19 were statistically associated with sex, marital status, education and occupation (p < 0.01). COVID-19 is currently the cause of death and it has a great impact on the economy, politics and social interactions in the study area. The government should strength the health system by increasing surveillance activities in detecting cases. Our findings suggest that the community should practice the WHO and EMoH recommendations to minimize the spread of the virus.

**Funding:** The authors received no specific funding for this work.

**Competing interests:** The authors have declared that no competing interests exist.

## 1. Introduction

Severe acute respiratory syndrome corona virus 2 (SARS-CoV-2) disease 2019 (COVID-19) was initially identified in China, Hubei Province, Wuhan city and declared as pandemic by WHO on the 31st December 2019 [1]. A study done in China reported that SARC-CoV-2 was obtained from lower respiratory tract samples identified from affected patients. The newly discovered virus was given the name COVID-19 in February 2020 [2]. This disease has become a major problem on public health across the world and creates a great fear within the people and spreads rapidly all over the world [3]. The consequences of the disease in terms of psychology, politics, economic and culture was beyond anything experienced in a generation [4]. SARS-CoV-2 is a single-stranded RNA new strain of virus that can infect humans and causes SARS-CoV-2 disease 2019 (COVID-19) and it mainly attacks the respiratory system of the body [5]. The virus is highly contagious and can easily be transmitted from human to human through respiratory droplets when infected people sneeze or cough and it has survival capacities on the surface of different materials for several hours [6]. The virus easily transmitted to the healthy person by touching droplet-contaminated surfaces or objects and then touching the eyes, nose or mouth [7]. The major clinical symptoms of COVID-19 are fever, cough, malaise, fatigue and difficulty to breathe [8]. The highly contagious nature of COVID-19 makes it to spread an alarming rate throughout the world, including Ethiopia [9, 10]. The severity of the disease is high in individuals of older age, diabetes, hypertension, cancer, kidney, lung and heart diseases [10, 11]. There are no scientifically produced proven antiviral drugs or vaccines on hand for the disease since now to control the spread of COVID-19 and thus create a significant risk to health care delivery [12]. Many countries throughout the world planned different strategies like staying at home, practicing social distancing, remote office activities, international travel bans, and using face masks to minimize the transmission of the virus [13, 14].

SARS-CoV-2 disease 2019 (COVID-19) was firstly identified in Ethiopia on March 15, 2020. The disease is rapidly expanding throughout the country starting from its appearance up to now. Following the first confirmation of the virus in the country, the Ethiopian government declared different restriction strategies to prevent the spreading of the disease which is in line with WHO [15]. According to Ethiopian Ministry of Health (EMoH) and the Ethiopian Public Health Institute (EPHI) report on October 22, 2020, there were 91,118 confirmed COVID-19 cases, 1,384 deaths and 44,506 recoveries in the country due to COVID-19. Addis Ababa, the capital city of Ethiopia, epicenter of COVID-19, is highly hit by COVID-19 [16]. Restriction of social activities to prevent the disease throughout the world causes a great impact on global economic fall [17, 18] and this is also a problem of Ethiopia which affects the socio-economic development and political environment. COVID-19 also affects the community daily life in different ways, such as physically, mentally and psychologically. This problem could be succeeded through knowledge, attitude, and practice (KAP) studies [19].

In Ethiopia, there is no doubt that health infrastructure is inadequate and needs considerable improvements. However, beyond the physical infrastructure of health centers, sociocultural and religious practices, public attitudes and behavior, pay no attention to practice of curfew, all precautionary measures and physical distancing are the factors that have a strong bearing on the spread of the virus and therefore require considerable attention in framing policy measures to tackle the crisis. The aim of this study is to investigate the community' knowledge, attitude and practices towards COVID-19 in Woldia town, Northeast Ethiopia. The result obtained from this study may help the government, the health workers and the community takes a better measurement to prevent COVID-19 pandemic.

## 2. Methods

### Description of the study area

This study was done in Woldia town, the main town of North Wollo, located 530km North East of Addis Ababa, the capital city of Ethiopia [Fig 1]. The town's geographical coordination is 11˚50′N latitude and 39˚36′E longitude. The town has an elevation of 2112 m above sea level.

### Study design

A cross-sectional survey study was done from July to September, 2020 in Woldia town, North-east Ethiopia.

### Study population

All the households that live in Woldia town who were aged 18 and above were included in this study.

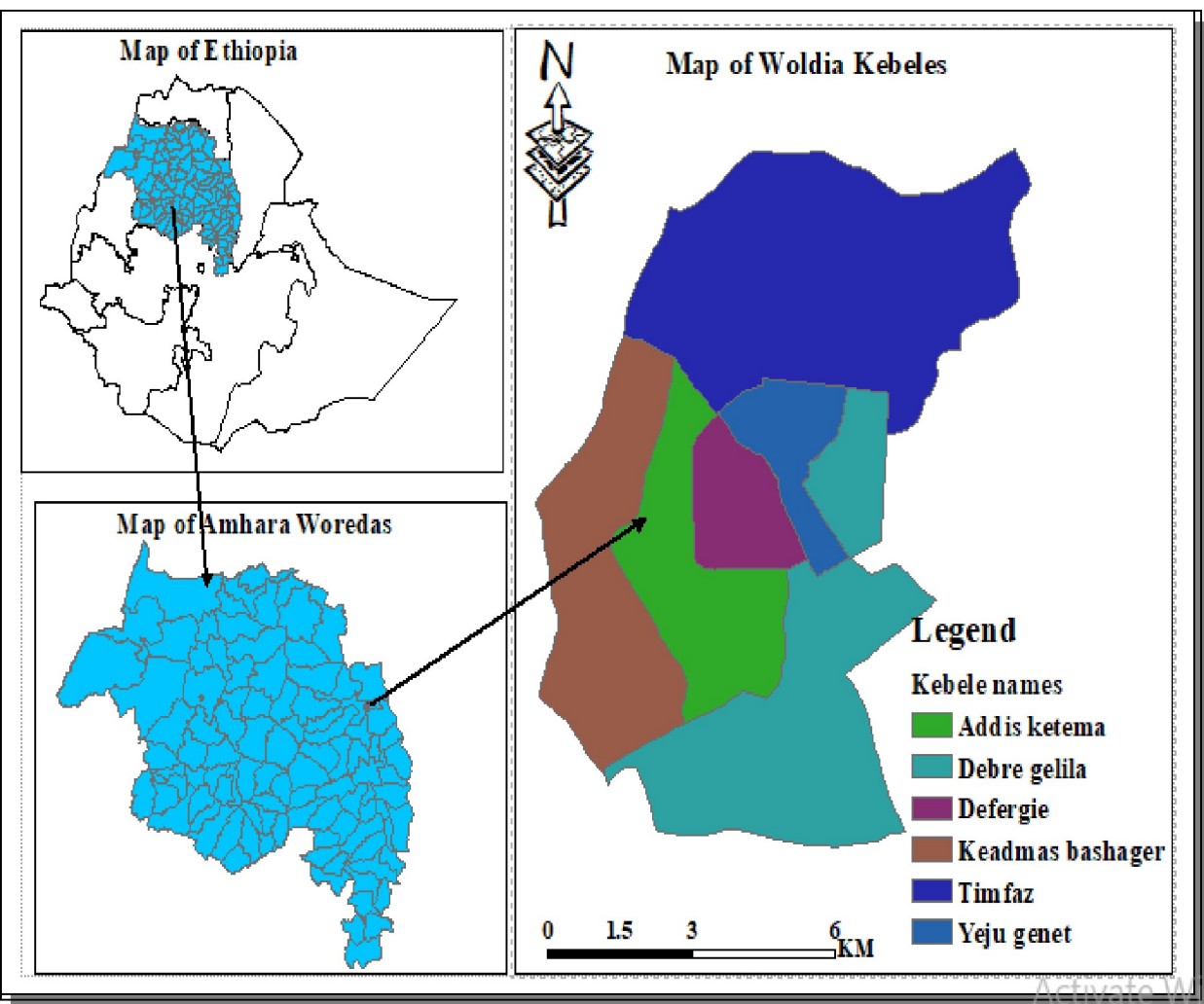

**Fig 1. Map of the study area.** Source: www.diva-gis.org/gdat.

## Ethical approval

The ethical committee of Woldia University approved our study protocol to collect the data from the respondents. The respondents' right to refuse or withdraw from participating was fully maintained and the information provided by each respondent was kept strictly confidential. After we gained willingness and written consent from the respondents, the required data was collected from the study participants.

## Sample size determination

To determine the sample size a single population proportion formula, $n = Z^2 p(1-p)/d^2$ was used. We used 50% of prevalence to get a representative sample size by considering 95% confidence interval, marginal error (d) of 5% and 5% non-response rate. Therefore, the minimum calculated sample size was 404. A random sampling technique was employed to select the study participants.

$n = Z^2 p(1-p)/d^2$ thus $n = (1.96)^2 (0.5)(1-0.5)/(0.05)^2 = 384 + 5\%$ non-response rate (19) = 404, where

n = Number of sample size

Z = 95% confidence interval equals 1.96

P = 0.5 (50%), proportion households (HHs) expected to practice in KAP

d = 0.05 (5%), marginal error

## Sampling technique and procedure

Based on geographical location of the town 6 small administrative units in a town ("Kebeles") of the town were divided into three strata. Secondly, from each stratum one "Kebele" was selected by random sampling technique. Thirdly, 404 households were selected proportionally from the three "Kebeles" based on their population size [Fig 2].

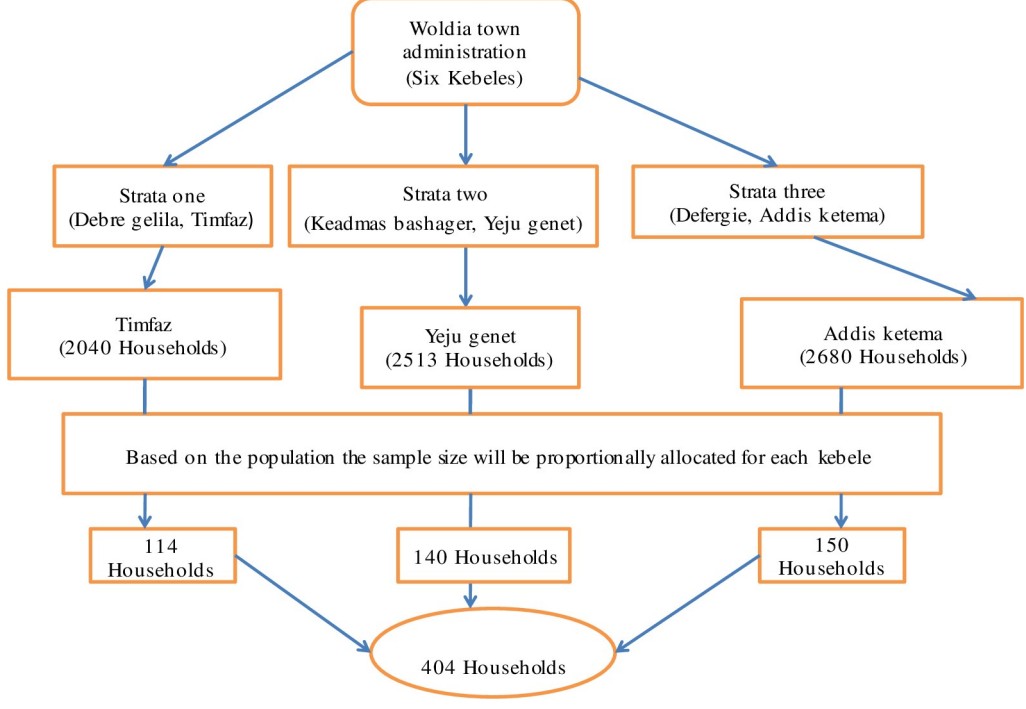

**Fig 2. Schematic representation of sampling procedure.**

## Study instrument

Semi-structured and interviewer-administered questionnaire was used to collect data from 404 respondents. The questionnaire to measure the respondents' knowledge, attitude and practice response about COVID-19 was adapted from previous research [10]. The semi-structured questionnaire includes socio-demographic characteristics including sex, age, marital status, education and occupation and KAP of the community towards COVID-19 prevention. The questionnaire was prepared in the English and Amharic languages. The data from the respondents were collected by wearing a face mask and glove at air ventilated area, keeping a minimum of 2m distance from the respondents. The questionnaire consisted three sections. The first section consisted information on respondents' socio-demographic characteristics. The second section contains participants' Knowledge, attitude and practices about COVID-19. The questionnaires assessing Knowledge, attitude and practice of the community towards COVID-19 were answered in yes/no. The third section assessed restriction strategy implementation by the community declared by the government towards COVID-19 by using a five-point Likert scale. For each of the five statements, respondents were asked to state their level of agreement, from "strongly agree," "agree," "undecided," "disagree," or "strongly disagree".

## Statistical analysis

In this study, the collected data were analyzed using the Statistical Package for the Social Sciences (SPSS), version 21. Descriptive analysis focused on frequency and percentage while chi-square test was utilized to describe statistical association between socio-demographic characteristics and KAP. A $p < 0.05$ was considered statistically associated.

## 3. Results

### Socio-demographic characteristics of respondents

To assess the community' knowledge, attitude and practices to SARS-CoV-2 disease 2019 (COVID-19) a total of 404 participants were involved in this study. The majority of the respondents (64.1%, n = 259/404) were in the age range of 18–39. In terms of sex, the majority participants were males. Regarding marital status (50.0%, n = 222/404) of the study participants were married. Out of the total study participants in terms of educational status, the majority (39.6%, n = 160/404) were diploma and above. Most (44.0%, n = 178/404) of the participants were employed [Table 1].

### Knowledge of respondents towards COVID-19

Regarding information about COVID-19 pandemic, the majority (76.5%, n = 309/404) of the study participants has got information from mass media (TV and radio). In terms of knowledge about the transmission mode of SARS-CoV-2 disease 2019 (COVID-19), the majority of the respondents knew that the virus can transmit through coughing and sneezing, direct contact with infected person and touching contaminated materials. The majority of the respondents (97.8%, n = 395/404), (97.5%, n = 394/404), (97.0%, n = 392/404) and (96.5%, n = 390/404) knew fever, dry cough, sneezing and difficulty to breathe are the clinical symptoms of the disease, respectively. Most (55.4%, n = 224/404) of the study participants responded that pseudoscientific practices (using garlic, ginger, feto, etc. as home remedies) are necessary to confront the SARS-CoV-2. The majority of the respondent (80.7%, n = 326/404) responded isolation and treatment of people who are infected with COVID-19 were effective ways to reduce the spread of viruses [Table 2]. The knowledge of the respondents about the

**Table 1. Demographic characteristics of the respondents towards COVID-19 (n = 404).**

| Variables | | Frequency (n = 404) | Percent (%) |
|---|---|---|---|
| **Age in years** | 18–39 | 259 | 64.1 |
| | > 39 | 145 | 35.9 |
| **Sex** | Male | 205 | 50.7 |
| | Female | 199 | 49.3 |
| **Marital status** | Single | 70 | 17.3 |
| | Married | 222 | 55.0 |
| | Divorced | 80 | 19.8 |
| | Widowed | 32 | 7.9 |
| **Education** | Illiterate | 16 | 4.0 |
| | Literate | 49 | 12.1 |
| | Primary | 146 | 36.1 |
| | Secondary | 33 | 8.2 |
| | Diploma and above | 160 | 39.6 |
| **Occupation** | Employed | 178 | 44.0 |
| | Unemployed | 130 | 32.2 |
| | Retired | 16 | 4.0 |
| | Business | 80 | 19.8 |

transmission mode of COVID-I9 through coughing and sneezing, direct contact with infected person and touching contaminated materials were statistically (p < 0.001) associated with education and occupation [Table 3].

**Table 2. Knowledge of the respondents towards COVID-19 (n = 404).**

| Variables | Frequency (n = 404) | Percent (%) |
|---|---|---|
| Source of knowledge about covid-19 | | |
| Mass media | 309 | 76.5 |
| Social media | 14 | 3.5 |
| Friends | 81 | 20.0 |
| Transmission mode | | |
| Through coughing and sneezing | 374 | 92.6 |
| Through direct contact with infected person | 366 | 90.6 |
| Through touching contaminated material | 329 | 81.4 |
| **Mean** | **368** | **91.2%** |
| Clinical symptoms of COVID-19 | | |
| Fever | 395 | 97.8 |
| Dry cough | 394 | 97.5 |
| Sneezing | 392 | 97.0 |
| Difficulty to breathe | 395 | 97.8 |
| Pseudoscientific practices (using garlic, ginger, feto, etc. as home remedies) are necessity to confront corona virus | | |
| Yes | 224 | 55.4 |
| No | 180 | 44.6 |
| Isolation and treatment of people who are infected with COVID-19 are effective ways to reduce the spread of viruses | | |
| Yes | 326 | 80.7 |
| No | 78 | 19.3 |

**Table 3. Bivariate analysis to determine the association of knowledge with socio- demographic characteristics of the respondents towards COVID-19 (n = 404).**

| | Knowledge, n (%) | | | | | |
| --- | --- | --- | --- | --- | --- | --- |
| Variables | Coughing and sneezing | | Direct contact with infected person | | Touching contaminated material | |
| | Yes | p-value | Yes | p-value | Yes | p-value |
| **Age in years** | | | | | | |
| 18–39 | 243(60.1) | 0.201 | 235(58.2) | 0.898 | 217(53.7) | 0.105 |
| >39 | 131(32.4) | | 131(32.4) | | 112(27.7) | |
| **Sex** | | | | | | |
| Male | 182(45.0) | < 0.001 | 175(43.3) | < 0.001 | 163(40.3) | 0.313 |
| Female | 192(47.5) | | 191(47.3) | | 166(41.1) | |
| **Marital status** | | | | | | |
| Single | 62(15.3) | | 62(15.3) | | 58(14.4) | |
| Married | 211(52.2) | 0.171 | 212(52.5) | < 0.001 | 175(43.3) | 0.386 |
| Divorced | 73(18.1) | | 65(16.1) | | 70(17.3) | |
| Widowed | 28(6.9) | | 27(6.7) | | 26(6.4) | |
| **Education** | | | | | | |
| Illiterate | 11(2.7) | | 11(2.7) | | 14(3.5) | |
| Literate | 37(9.2) | < 0.001 | 41(10.1) | < 0.001 | 36(8.9) | 0.070 |
| Primary | 139(34.4) | | 140(34.7) | | 115(28.5) | |
| Secondary | 31(7.7) | | 29(7.2) | | 24(5.9) | |
| Diploma and above | 156(38.5) | | 145(35.9) | | 140(34.7) | |
| **Occupation** | | | | | | |
| Employed | 169(41.8) | | 164(40.6) | | 152(37.6) | |
| Unemployed | 125(30.9) | < 0.001 | 119(29.5) | < 0.001 | 99(24.5) | 0.192 |
| Retired | 9(2.2) | | 8(2) | | 12(3) | |
| Business | 71(17.6) | | 75(18.6) | | 66(16.3) | |

## Attitude of the participants towards COVID-19

This study showed that (46.3%, n = 187/404) of the participants had a positive attitude in the control of COVID-19 pandemic. The majority (51.0%, n = 198/404) of the respondents had a negative attitude that the government has not taken sufficient effective measures to prevent the spread of COVID-19. This study revealed that (55.9%, n = 226/404) of the study participants were not in doubt that breast feeding cannot transmit COVID-19 from mother to baby. Similarly (59.7%, n = 241/404) of the participants were confident that children are not susceptible to COVID-19 pandemic [Table 4]. The attitudes of the respondents towards successfully controlling of COVID-I9 was statistically (p < 0.001) associated with age, marital status, education and occupation [Table 5].

## Practices of the respondents about COVID-19

Vast majority of the surveyed participants of this study (63.1%, n = 255/404) didn't frequently wash their hands with soap. Similarly (72.8%, n = 294/404) didn't frequently clean their hands with sanitizer. Most (63.6%, n = 257/404) of the surveyed participants had a good practice of covering their mouth and nose while coughing. Regarding of avoidance of touching mouth and nose (58.9%, n = 238/404) of the participants responded that they did not avoid touching mouth and nose. Among the study participants (66.8%, n = 270/404) were not practicing social distancing. Most of the respondents (63.9%, n = 258/404) haven't worn face masks when they go in public or crowded places. Among the total respondents (67.8%, n = 274/404) were not

**Table 4. Attitude of the participants towards COVID-19 (n = 404).**

| Variables | Frequency (n = 404) | Percent (%) |
|---|---|---|
| COVID-19 will be successfully controlled | | |
| Yes | 187 | 46.3 |
| No | 217 | 53.7 |
| The government has taken sufficient effective measures to prevent the spread of COVID-19 | | |
| Yes | 206 | 49.0 |
| No | 198 | 51.0 |
| COVID-19 can transfer from mother to babies through breast feeding | | |
| Yes | 178 | 44.1 |
| No | 226 | 55.9 |
| Children are not susceptible to COVID-19 | | |
| Yes | 241 | 59.7 |
| No | 163 | 40.3 |

cleaning their working place surface [Table 6]. The practices of the respondents towards COVID-19 were statistically (p < 0.01) associated with sex, marital status, education and occupation [Table 7].

**Table 5. Bivariate analysis to determine the association of attitude with socio-demographic characteristics of the respondents towards COVID-19 (n = 404).**

| Variables | Attitudes, n (%) | | |
|---|---|---|---|
| | Covid-19 will be successfully controlled | | |
| | Yes | No | p-Value |
| **Age** | | | |
| 18–39 | 105(26) | 154(38.1) | < 0.001 |
| > 39 | 82(20.3) | 63(15.6) | |
| **Sex** | | | |
| Male | 101(25) | 104(25.7) | 0.223 |
| Female | 86(21.3) | 113(28) | |
| **Marital status** | | | |
| Single | 40(9.9) | 30(7.4) | < 0.001 |
| Married | 99(24.5) | 123(30.4) | |
| Divorced | 44(10.9) | 36(8.9) | |
| Widow | 4(1.0) | 28(6.9) | |
| **Education** | | | |
| Illiterate | 2(0.5) | 14(3.5) | < 0.001 |
| Literate | 27(6.7) | 22(5.4) | |
| Primary | 103(25.5) | 43(10.6) | |
| Secondary | 7(1.7) | 26(6.4) | |
| Diploma and above | 49(12.1) | 111(27.5) | |
| **Occupation** | | | |
| Employed | 65(16.1) | 113(28) | < 0.001 |
| Unemployed | 55(13.6) | 75(18.6) | |
| Retired | 11(2.7) | 5(1.2) | |
| Business | 53(13.1) | 27(6.7) | |

Table 6. **Practices of the respondents about COVID-19 (n = 404).**

| Variables | Frequency (n = 404) | Percent (%) |
|---|---|---|
| Frequent hand washing with soap | | |
| Yes | 149 | 36.9 |
| No | 255 | 63.1 |
| Frequent hand washing with sanitizer | | |
| Yes | 110 | 27.2 |
| No | 294 | 72.8 |
| Covering mouth and nose while coughing | | |
| Yes | 257 | 63.6 |
| No | 147 | 36.4 |
| Avoidance of touching nose and mouth | | |
| Yes | 166 | 41.1 |
| No | 238 | 58.9 |
| Practicing of social distancing | | |
| Yes | 134 | 33.2 |
| No | 270 | 66.8 |
| Wearing of face masks in public or crowded places | | |
| Yes | 146 | 36.1 |
| No | 258 | 63.9 |
| Frequent cleaning of workplace surfaces | | |
| Yes | 130 | 32.2 |
| No | 274 | 67.8 |

## Government restriction strategies to prevent COVID-19

The Ethiopian government has declared different strategies to prevent the spread of COVID-19 in the community. Among the respondents (80.0%, n = 323/404) strongly disagreed about the implementation of restrictive strategy by staying home to prevent COVID-19. Table 8 revealed that (26.7%, n = 108/404) of the study participants agreed weeding ceremonies were implemented in the presence of overcrowded people. Among the total participants (56.0%, n = 226/404) of them strongly agreed that the community practice funeral ceremonies in the presence of crowded people. Similarly (52.0%, n = 210/404) of the study participants strongly agreed that the communities practice holiday activities without keeping social distancing. Most of the respondents (68.0%, n = 275/404) strongly agreed that gathering of people in religious places (churches, mosques) was common. Regarding to reduction of the number of public transporting passengers to 50% strategy (44.0%, n = 178/404) were strongly disagreed about its implementation to prevent the spread of the virus. Majority (44.0%, n = 178/404) of the respondents strongly disagreed that the communities regularly use face masks when they go to people crowded places (e.g., market, bus station) [Table 8].

## 4. Discussion

Currently, SARS-CoV-2 disease 2019 (COVID-19) is a global health problem and it creates fear within the people [20]. At present, even if there are trials to develop the vaccine, there is no effective vaccine and treatment for the SARS-CoV-2 disease 2019 (COVID-19) pandemic in the world. Therefore, the only solution to prevent the disease is increasing the communities' awareness, attitude and practices about the COVID-19 [9]. The aim of this study was to assess the communities' knowledge, attitude, practices and government restriction strategy implemented by the communities to prevent COVID-19 in Woldia town, Northeast Ethiopia. The

**Table 7. Bivariate analysis to determine the association of practices with socio-demographic characteristics of the respondents towards COVID-19 (n = 404).**

| Variables | Practices n (%) | | | | | | | |
|---|---|---|---|---|---|---|---|---|
| | Frequent hand washing with soap | | Avoiding touching nose and mouth | | Practicing of social distancing | | Wearing of facemasks | |
| | Yes | p-value | Yes | p-value | Yes | p-value | Yes | p-value |
| **Age** | | | | | | | | |
| 18–39 | 67(16.6) | < 0.001 | 73(18.1) | < 0.001 | 53(13.1) | < 0.001 | 89(22) | 0.321 |
| >39 | 80(19.8) | | 92(22.8) | | 80(19.8) | | 57(14.1) | |
| **Sex** | | | | | | | | |
| Male | 64(15.8) | < 0.05 | 108(26.7) | | 85(21) | < 0.001 | 79(19.6) | 0.351 |
| Female | 83(20.5) | | 62(15.4) | | 48 (11.9) | | 67(16.6) | |
| **Marital status** | | | | | | | | |
| Single | 1(0.2) | | 32(7.9) | | 17(4.2) | | 18(4.5) | |
| Married | 95(23.5) | < 0.001 | 105(26.0) | < 0.001 | 84(20.8) | < 0.001 | 74(18.3) | < 0.05 |
| Divorced | 50(12.4) | | 28(6.9) | | 32(7.9) | | 38(9.4) | |
| Widowed | 1(0.2) | | 1(0.2) | | 1(0.2) | | 16(4) | |
| **Education** | | | | | | | | |
| Illiterate | 2(0.5) | | 11(2.7) | | 5(1.2) | | 6(1.5) | |
| Literate | 14(3.5) | < 0.01 | 29(7.2) | < 0.001 | 17(4.2) | < 0.001 | 8(2.0) | < 0.001 |
| Primary | 69(17.1) | | 64(15.8) | | 64(15.8) | | 50(12.4) | |
| Secondary | 13(3.2) | | 14(3.5) | | 16(4.0) | | 2(0.5) | |
| Higher | 51(12.6) | | 48(11.9) | | 32(7.9) | | 80(19.8) | |
| **Occupation** | | | | | | | | |
| Employed | 52(12.9) | | 65(16.1) | | 33(8.2) | | 81(20) | |
| Unemployed | 48(11.9) | < 0.001 | 43(10.6) | < 0.001 | 37(9.2) | < 0.001 | 38(9.4) | < 0.01 |
| Retired | 13(3.2) | | 12(3.0) | | 16(4.0) | | 7(1.7) | |
| Business | 36(8.9) | | 46(11.4) | | 48(11.9) | | 20(5.0) | |

majority of the study participants were male and in the age range of 18–39. The finding of this study is in line with the findings done in Africa [21] where 83.3% were between 18 to 39 years. The majority of the respondents (76.0%) got the information about COVID-19 from mass media such as TV and radio which was in agreement with the findings done in Addis Zemen Hospital, Ethiopia [10] and Egypt [22]. Most of the respondents had good knowledge about the transmission mode of COVID-19 that is in agreement with previous studies done in Kenya and Nigeria [8, 23]. Similarly, a study done in Asia indicated that the majority population had a very good knowledge about COVID-19 [24]. Of the respondents surveyed in this study (80.4%) had very good knowledge about the mode of transmission and the clinical symptoms of COVID-19. A previous study reported in Saudi Arabia [13, 25] is persistent with the findings of the present study. The good awareness of respondents about COVID-19 transmission ways and clinical symptoms was obtained through television and radio. Both the government mass media and non-government mass media had a great contribution to increase the awareness of the community towards COVID-19 pandemic. Our findings showed that (44.6%) of the participants responded pseudoscientific practices (using garlic, ginger, feto, etc. as home remedies) are not necessary to confront the SARS-CoV-2. Of the total respondents (80.7%) answered isolation and treatment of people who are infected with COVID-19 are effective ways to reduce the spread of viruses.

In the present study (46.3%) of the study participants had positive attitudes towards successful control of COVID-19 pandemic by the Ethiopian government. The finding of this study is not in agreement with the study done in Malaysia (83.1%) [19]. A study reported in

**Table 8. Community's practices to implement government restriction strategies to prevent COVID-19 (n = 404).**

| Variables | | Strongly agree No (%) | | Agree No (%) | | Undecided No (%) | | Disagree No (%) | | Strongly disagree No (%) |
|---|---|---|---|---|---|---|---|---|---|---|
| The communities implement the restriction strategy by staying home | M | - | M | 18(4.5%) | M | - | M | 22(5.5%) | M | 165 (40.9%) |
| | F | - | F | 15(3.5%) | F | - | F | 26(6.5%) | F | 158 (39.1%) |
| | **T** | **-** | **T** | **33(8%)** | **T** | **-** | **T** | **48(12%)** | **T** | **323(80%)** |
| Weeding ceremonies are implemented in the presence of overcrowded people | M | 52(12.8%) | M | 34(8%) | M | 39(9.5%) | M | 40(10%) | M | 33(8.1%) |
| | F | 56(13.9%) | F | 31(7.5%) | F | 42(10%) | F | 45(11%) | F | 32(7.9%) |
| | **T** | **108 (26.7%)** | **T** | **65 (15.5%)** | **T** | **81 (19.5%)** | **T** | **85(21%)** | **T** | **65(16%)** |
| Funeral ceremonies are implemented in the presence of crowded people | M | 116(29%) | M | 24(5.9%) | M | 17(4.3%) | M | 8(2%) | M | 39(9.5%) |
| | F | 110(27%) | F | 25(6.1%) | F | 15(3.7%) | F | 8(2%) | F | 42(10.5%) |
| | **T** | **226(56%)** | **T** | **49(12%)** | **T** | **32(8%)** | **T** | **16(4%)** | **T** | **81(20%)** |
| Peoples celebrate holidays without implementing social distancing. | M | 107 (26.5%) | M | 33(8.1%) | M | 15(3.7%) | M | 20(5%) | M | 23(5.8%) |
| | F | 103 (25.5%) | F | 32(7.9%) | F | 17(4.3%) | F | 29(7%) | F | 25(6.2%) |
| | **T** | **210(52%)** | **T** | **65(16%)** | **T** | **32(8%)** | **T** | **49(12%)** | **T** | **48(12%)** |
| Gathering of people in religious places (churches, mosques) is common | M | 136 (33.6%) | M | 31(7.6%) | M | 7(1.8%) | M | 8(2%) | M | 9(2.2%) |
| | F | 139 (34.4%) | F | 34(8.4%) | F | 9(2.2%) | F | 8(2%) | F | 7(1.8%) |
| | **T** | **275(68%)** | **T** | **65(16%)** | **T** | **16(4%)** | **T** | **16(4%)** | **T** | **16(4%)** |
| Reduction of the number of public transport passengers to 50% is well implemented | M | 52(12.8%) | M | 40(9.9%) | M | 18(4.5%) | M | 9(2.2%) | M | 91(22.5%) |
| | F | 45(11.2%) | F | 41 (10.1%) | F | 14(3.5%) | F | 7(1.8%) | F | 87(21.5%) |
| | **T** | **97(24%)** | **T** | **81(20%)** | **T** | **32(8%)** | **T** | **16(4%)** | **T** | **178(44%)** |
| The communities regularly use face masks when they go to people crowded places (e.g., market, bus station) | M | 17(4.3%) | M | 25(6.1%) | M | 9(2.2%) | M | 60 (14.9%) | M | 93(23%) |
| | F | 15(3.7%) | F | 24(5.9%) | F | 7(1.8%) | F | 69 (17.1%) | F | 85(21%) |
| | **T** | **32(8%)** | **T** | **49(12%)** | **T** | **16(4%)** | **T** | **129 (32%)** | **T** | **178(44%)** |

M: Male; F: Female; T: Total.

china [24] the majority of the respondents had a positive attitude that COVID-19 would be able to control successfully. In this study (36.9%) and (27.22%) of the respondents frequently wash their hands with soap and sanitizers, respectively. A similar study done in Ecuador [26] reported that the majority (96.6%) of the respondents was washed their hands for at least 20 seconds every time after returning house or touching another person which was not in persistent with the present study. This indicates the practicing habit of the respondents were very poor. A similar study done in Pakistan (88.1%) is in not in line with our findings [27]. Another study done in Nigeria (89.0%) of the respondents responded they frequently washed their hands with soap and water or hand sanitizer [28] which is not in agreement with the present study. However, (58.9%) of the study participants did not avoid touching the nose and mouth with hands, which is inconsistent with a study done in Addis Zemen Hospital, Ethiopia [10] where the majority (74%) of the respondents was not practicing of touching their nose and mouth with their hands. In the present study (66.8%) of the study participants did not practice

social distancing to prevent the spread of COVID-19 pandemic. However, in Indonesia the majority (93%) of the respondents had a very good practice towards social distancing for preventing the outbreak of COVID-19 [29]. difference might be due to educational status that the majority of the respondents in the present study were illiterate, literate, primary and secondary school completion. Of the total respondents (80.0%) said that the communities were obeying the government restriction strategy of staying at home to prevent COVID-19 pandemic. Another study done in Nigeria and Egypt [21] only (36.0%) of the participants practice health recommendations during the lockdown period. The reason of low practice to stay home might be due to socio-economic problems faced by the community. Our findings revealed (26.7%) and (56.0%) of the respondents were obeying the government restriction strategies of avoiding weeding ceremonies and funeral ceremonies, respectively, in the presence of crowded people without practicing social distancing. This might be due to lack of strict enforcement of the required lockdown. However, a similar study done in China, a majority of the participants (96.4%) avoided attending in people crowded places [8]. The reason for this difference might be due to socioeconomic, religious and educational differences between the study populations. The community also has a wrong attitude and believed that the government use for political consumption. In the current study (44%) of the study participants responded that the community in the study area did not wear face masks when they go to people crowded places (e.g., market, bus station). This is in line with a study has done in Addis Zemen Hospital, Ethiopia [10] but in contrary to the findings has done in China [8] only (4.0%) went to crowded areas or went outside without a face mask. The low practice of wearing face masks in the present study might be due to low income to buy it and shortage of the masks in the study area. Similarly, the low commitment of the local government to enforce the community also might be a reason for poor practice to wear face masks in the study area.

## 5. Limitations

In this study one of the limitation was the respondents might give socially desirable responses. The other limitation of this study was we could not compare the knowledge, attitude and practices of the respondents between age groups, sex, education, marital status and occupation. Additional limitation of the present study was related to the standardization of tools we used to assess the knowledge, attitude and practices of the respondents. The instrument was adapted from a survey that had been previously tested and utilized in Ethiopia [10]. Therefore, the mentioned factors are limitations to the representativeness of the results.

## 6. Conclusion

The present study assessed the community knowledge, attitude and practices to SARS-CoV-2 disease 2019 (COVID-19) in Woldia town, Northeast Ethiopia. In our study, the majority (91.2%) of the study participants had good prevention knowledge towards the COVID-19 pandemic. However, the practical knowledge and implementation of governmental restriction strategies to prevent the disease is very low. Only 36.9%, 36.1% and 33.2% of the participants practice hand washing with soap, wear face masks and social distancing, respectively. The majority of the community is lacking safety practices like celebrating holidays with crowded people, wearing face masks and social distancing which have a great contribution of spread of the disease. We recommend that implementing frequent washing of hands with soap and sanitizer, practicing social distancing and face mask wearing could inhibit the spread of COVID-19 and successful control in the study area.

## Acknowledgments

We would like to thank all the respondents involved in the interview for kindly given us the necessary data.

## Author Contributions

**Conceptualization:** Kindu Alem Molla, Silamlak Birhanu Abegaz.

**Data curation:** Kindu Alem Molla, Silamlak Birhanu Abegaz.

**Methodology:** Kindu Alem Molla, Silamlak Birhanu Abegaz.

**Resources:** Kindu Alem Molla.

**Writing – original draft:** Kindu Alem Molla.

**Writing – review & editing:** Kindu Alem Molla, Silamlak Birhanu Abegaz.

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
