## [Decision Letter · Decision Letter 0]

4 Jan 2021

PONE-D-20-35696

Community Knowledge, Attitude and Restriction Practices on Novel Corona Virus Disease (COVID-19) in Woldia town, Northeast Ethiopia

PLOS ONE

Dear Dr.Molla

Thank you for submitting your manuscript to PLOS ONE. After careful consideration, we feel that it has merit but does not fully meet PLOS ONE’s publication criteria as it currently stands. Therefore, we invite you to submit a revised version of the manuscript that addresses the points raised during the review process.

We look forward to receiving your revised manuscript.

Kind regards,

Amit Sapra

Academic Editor

PLOS ONE

2. Please note that whilst COVID-19 refers to the name of the disease, the name of the virus that cause it is SARS-CoV-2.

4. We note that Figure 1 in your submission contains map images which may be copyrighted. All PLOS content is published under the Creative Commons Attribution License (CC BY 4.0), which means that the manuscript, images, and Supporting Information files will be freely available online, and any third party is permitted to access, download, copy, distribute, and use these materials in any way, even commercially, with proper attribution. For these reasons, we cannot publish previously copyrighted maps or satellite images created using proprietary data, such as Google software (Google Maps, Street View, and Earth). For more information, see our copyright guidelines: http://journals.plos.org/plosone/s/licenses-and-copyright.

(1) You may seek permission from the original copyright holder of Figure 1 to publish the content specifically under the CC BY 4.0 license. 

5. Please ensure you have discussed any potential limitations of your study in the Discussion, including study design, sample size and/or potential confounders.

6. In your Methods section, please provide additional information about the participant recruitment method and the demographic details of your participants. Please ensure you have provided sufficient details to replicate the analyses such as a description of how participants were recruited.

7. Please include additional information regarding the survey or questionnaire used in the study and ensure that you have provided sufficient details that others could replicate the analyses. For instance, if you developed a questionnaire as part of this study and it is not under a copyright more restrictive than CC-BY, please include a copy, in both the original language and English, as Supporting Information.

8. Thank you for including your ethics statement: "Ethical approval clearance was obtained from ethics research committee of Woldia University."   

9. Please provide additional details regarding participant consent. In the ethics statement in the Methods and online submission information, please ensure that you have specified 1) whether consent was informed, and 2) what type you obtained (for instance, written or verbal, and if verbal, how it was documented and witnessed).

10. Thank you for stating the following financial disclosure:

"No funding for this work." 

11. Thank you for stating the following in your Competing Interests section: 

"NO authors have competing interests"

12. Please ensure that you refer to Figure 2 in your text as, if accepted, production will need this reference to link the reader to the figure.

Reviewers' comments:

Reviewer's Responses to Questions

**Comments to the Author**

1. Is the manuscript technically sound, and do the data support the conclusions?

Reviewer #1: Partly

2. Has the statistical analysis been performed appropriately and rigorously? 

Reviewer #1: I Don't Know

3. Have the authors made all data underlying the findings in their manuscript fully available?

Reviewer #1: Yes

4. Is the manuscript presented in an intelligible fashion and written in standard English?

Reviewer #1: Yes

5. Review Comments to the Author

Reviewer #1: 1) Language is unclear and difficult to follow. The article needs thorough evaluation by a copy editor to improve readability.

2) I am not sure that the mention made by authors that there is no scientifically vaccines holds true since a lot of vaccine candidates are in late phase trials already.

3) I request authors to explain what does the term " Kebeles" .

4) The abstract needs to be written completely as has incomplete sentences . The conclusion of the abstract does not seem appropriate.

5) Authors have mentioned the Novel Corona Virus as " Nobel" Corona virus at multiple places.

6) The data presented by the authors describes about the socio-demographics of the study population but does not elaborate how the KAP are correlated. The authors should elaborate on how the KAP are being affected by the education, occupation or age of the study population.

7) The authors have compared their study with other studies in other countries but comparisons need to be elaborated upon and need data from other studies as well. For example the authors mention - The findings of this study are not in agreement with the study done in Malaysia. No data is provided about the Malaysian study.

6. PLOS authors have the option to publish the peer review history of their article (what does this mean?). If published, this will include your full peer review and any attached files.

Reviewer #1: No

---

## [Author Response · Author response to Decision Letter 0]

16 Feb 2021

The responses for the comments provided both by the Editor and reviewer are included the corrected manuscript.

---

## [Decision Letter · Decision Letter 1]

29 Mar 2021

PONE-D-20-35696R1

Community knowledge, attitude and practices to SARS-CoV-2 disease 2019 (COVID-19): A cross-sectional study in Woldia town, Northeast Ethiopia

PLOS ONE

Dear Dr. Molla,

Thank you for submitting your manuscript to PLOS ONE. After careful consideration, we feel that it has merit but does not fully meet PLOS ONE’s publication criteria as it currently stands. Therefore, we invite you to submit a revised version of the manuscript that addresses the points raised during the review process.

We look forward to receiving your revised manuscript.

Kind regards,

Wen-Jun Tu

Academic Editor

PLOS ONE

Journal Requirements:

Additional Editor Comments (if provided):

1. In order to provide a more complete information to our readers on the topic, we would like to emphasize the importance to cross referencing very recent material on the same topic published in "PLoS ONE ". Therefore, it would be highly appreciated if you would check the contents published in the last two years of "PLoS ONE" (https://journals.plos.org/plosone/) and add all material relevant to your article to the reference list.

2. add“Clinical Features and Short-term Outcomes of 102 Patients with Corona Virus Disease 2019 in Wuhan, China. Clinical Infectious Diseases, 71(15):748-755” in the revision text

Reviewers' comments:

Reviewer's Responses to Questions

**Comments to the Author**

1. If the authors have adequately addressed your comments raised in a previous round of review and you feel that this manuscript is now acceptable for publication, you may indicate that here to bypass the “Comments to the Author” section, enter your conflict of interest statement in the “Confidential to Editor” section, and submit your "Accept" recommendation.

Reviewer #1: All comments have been addressed

2. Is the manuscript technically sound, and do the data support the conclusions?

Reviewer #1: Yes

3. Has the statistical analysis been performed appropriately and rigorously? 

Reviewer #1: Yes

4. Have the authors made all data underlying the findings in their manuscript fully available?

Reviewer #1: Yes

5. Is the manuscript presented in an intelligible fashion and written in standard English?

Reviewer #1: Yes

6. Review Comments to the Author

Reviewer #1: (No Response)

7. PLOS authors have the option to publish the peer review history of their article (what does this mean?). If published, this will include your full peer review and any attached files.

Reviewer #1: **Yes: **Priyanka Bhandari MD

---

## [Author Response · Author response to Decision Letter 1]

2 Apr 2021

The authors have tried to include the comments in the revised version of the manuscript that addresses the points raised during the review process.

---

## [Editor Report · Decision Letter 2]

7 Apr 2021

Community knowledge, attitude and practices to SARS-CoV-2 disease 2019 (COVID-19): A cross-sectional study in Woldia town, Northeast Ethiopia

PONE-D-20-35696R2

Dear Dr. Molla,

We’re pleased to inform you that your manuscript has been judged scientifically suitable for publication and will be formally accepted for publication once it meets all outstanding technical requirements.

Kind regards,

Wen-Jun Tu

Academic Editor

PLOS ONE
---

## [Editor Report · Acceptance letter]

12 Apr 2021

PONE-D-20-35696R2 

Community knowledge, attitude and practices to SARS-CoV-2 disease 2019 (COVID-19): A cross-sectional study in Woldia town, Northeast Ethiopia 

Dear Dr. Molla:

I'm pleased to inform you that your manuscript has been deemed suitable for publication in PLOS ONE. Congratulations! Your manuscript is now with our production department. 

Kind regards, 

on behalf of

Dr. Wen-Jun Tu 

Academic Editor

PLOS ONE